# Structural and Physicochemical Properties of Tunisian *Quercus suber* L. Starches for Custard Formulation: A Comparative Study

**DOI:** 10.3390/polym14030556

**Published:** 2022-01-29

**Authors:** Youkabed Zarroug, Mouna Boulares, Dorra Sfayhi, Bechir Slimi, Bouthaina Stiti, Kamel Zaieni, Sirine Nefissi, Mohamed Kharrat

**Affiliations:** 1Field Crops Laboratory (LR20INRAT02), National Agricultural Research Institute of Tunisia (INRAT), University of Carthage, Ariana 2049, Tunisia; sfayhi.dorra@yahoo.fr (D.S.); mohamed_kharrat@outlook.com (M.K.); 2Research Unit: Bio-Preservation and Valorization of Agricultural Products (UR13-AGR 02), Higher Institute of Food Industries (ESIAT), University of Carthage, Tunis 1003, Tunisia; boulares_mouna2006@yahoo.fr (M.B.); nefisi.sirine@gmail.com (S.N.); 3Laboratoire des Nanomatériaux et Systèmes Pour les Énergies Renouvelables (LANSER), Centre de Recherches et des Technologies de l’Energie Technopole Borj Cedria, BT 95, Hammam Lif 2050, Tunisia; bechir.slimi0511@gmail.com (B.S.); kamelzayani.71@gmail.com (K.Z.); 4National Institute of Research in Rural Engineering, Water and Forests BP 10, University of Carthage, Ariana 2080, Tunisia; stitibou@gmail.com

**Keywords:** acorn, starch, structure, technological properties, custard, characterization

## Abstract

The aim of the present study was to extract starch from acorn (*Quercus suber* L.) fruits using water and alkaline methods. Structural and functional properties of extracted starches were investigated and compared to those of corn and modified starches in order to determine their innovative potential application in food industry. The yield of extraction using the two methods was about 48.32% and 48.1%. The isolated starches showed low moisture, fat and protein contents, revealing high purity and quality. Additionally, the starch extracted using the alkaline method (AAS) showed higher lightness (60.41) when compared to starch isolated using hot water (WAS). However, the lightest white color was found for studied commercial starches. Moreover, AAS starch exhibited the highest swelling power, solubility and water absorption, followed by WAS and commercial starches. Results showed that extracted acorn starches were characterized by greater enthalpy and gelatinization temperatures. Similar observations were noted using FT-IR spectra analysis for all analyzed starches. In addition, granule starches observed using scanning electron microscopy were found to be spherical and ovoid. However, from the analysis by X-ray diffraction, a crystalline pattern of C-type was found for acorn starches, while commercial starches presented an A-type pattern. As an innovative food application, these underexploited acorn starches were valued and served to produce new custards with improved functional properties and better microstructure when compared to commercial custard.

## 1. Introduction

The genus *Quercus*, which belongs to the Fagaceae family, contains around 600 species worldwide [1] such as *Quercus suber*, *Quercus robur*, *Quercus petraea*, *Quercus ilex*, and *Quercus pubescens* [2]. The acorn fruit is widespread in the Mediterranean forests in North Africa, especially in Algeria, Morocco, and Tunisia. Traditionally, several acorn plants have been widely used in Mediterranean medicine and foods. Due to their high concentration of tannic acid, a mild toxin giving them a bitter taste, the acorn fruits are soaked in water, boiled, or roasted and then consumed in the form of flour or grilled as a substitute for coffee beans. In Tunisia, *Quercus suber* trees occupy almost 70,000 ha and predominantly grow in “Ain Drahem” and Tabarka [3]. Recent studies showed that acorn fruits contain several bioactive compounds such as polyphenols, sterols, and long-chain alcohols [4,5] with interesting antioxidant activity [2]. In addition, *Quercus* fruits contain many valuable compounds such as fibers, proteins, minerals, and oil, as well as starch, constituting over 58% of the kernel [6]. Starch is the main reserve nutrient in plants, providing 70 to 80% of the world’s human caloric consumption [5,6,7]. It is a biodegradable and renewable polysaccharide, which consists of two glucopyranose homopolymers: amylose and amylopectin [8]. Chemically, amylose has a linear structure consisting of D-glucopyranosyl subunits attached by α-(1→4) glycoside linkages with the presence of slightly branched molecules. However, amylopectin is a broad branched polymer composed of α-(1→4) -D-glucopyranose units linked by α-(1→6) bonds at the branching points [9,10]. The most widely used starches have been extracted from many vegetable sources, especially from cereals, legumes, roots, and tuber crops [10]. To enhance the consumption of natural resources, many other new sources of starch are being studied, such as biri, ginger, arrowroot, sweet potato and yam [11], and maca or Peruvian carrot [12]. These native starches of different botanical origins reveal enormous diversity in their structure and composition, including their amylase, amylopectin, and minor constituent (lipid, protein, and mineral) contents. Starch is rarely consumed in its intact form and frequently used by industry in its native form. Most native starches are limited in their direct application because they are unstable with respect to changes in temperature, pH, and shear forces. Therefore, native starches are often modified to enhance their technological and functional properties under the high heating temperatures used in industrial processes. In order to replace the chemically modified starches, particularly in the food industry, researchers and nutritionists are searching for new natural starch sources such as acorn fruits [6,7,8,9,10,11,12,13]. Recently, a few studies have reported that acorn starch has received some research interest in the food (e.g., cereals, snacks, dairy, soups, jelly) and nonfood industries [14], especially in the development of biomaterials (biofilms) [15]. In the food industry, starch is used particularly to improve the properties of baking flour and bakery products such as cakes and breads due to their high consumption. During their formulations, starch is one of the components responsible for broad technological functionality. Generally, starch is used as a food additive, hydrocolloid, thickener, emulsifier, and coating agent [16] in many industrial foods to improve their functional and sensory characteristics. For the acorn extraction starch, many chemical-based technologies have been extensively applied in research, such as alcohols-based extraction, alkaline-based extraction and acetone-based extraction [17,18,19]. However, Zhang et al. [20] studied the effect of both physical and chemical methods, including hot-water soaking, alkaline washing, ultrasonic-assisted ethanol soaking, and ultrasonic-assisted hot-water soaking, on the extracted acorn starch yield. Compared to other extracted starches from cassava, barley, potato, and wheat, little research has been devoted to the evaluation of the structural and functional characteristics of acorn starch [21]. Acorns are non-conventional and good sources of a new form of starch that was not valued until now in food industry. In fact, few studies have focused on the functional properties of acorn starch from *Quercus* species without any food applications. The first application of acorn starch was studied in the investigation by Zarroug et al. [13], who incorporated acorn starch during the production of a fermented dairy product to enhance its functional properties. To our knowledge, the use of acorn starch for custard formulation has not been reported before, which shows the importance of this innovative application aiming to add value to this underexploited raw material in the food industry. The objective of this study was, first, to extract native acorn starches using two different methods (water and alkaline extraction) and compare their structural and physicochemical properties with those of commercial corn starch and modified starch. Second, the study aimed to investigate selected properties of a newly formulated acorn-based custard to provide information on the potential incorporation of acorn starch into various industrial food applications.

## 2. Materials and Methods

### 2.1. Materials

Acorn fruits belonging to the species *Quercus Suber* L. were manually collected from Beni M’tir region of Jendouba in the northwestern Tunisia. Acorn kernels were hand-peeled, dried at 40 °C for 3 days, and then milled into flour. The acorn flour was used to extract starch by two extraction methods: water extraction, as reported in the study by Singh and Singh [22], and alkaline extraction as described in the previous study by Zarroug et al. [13]. The starches extracted with the water and alkaline methods were named water acorn starch (WAS) and alkaline acorn starch (AAS), respectively. The obtained starches were dried at 40 °C for 12 h and milled into a fine powder. Then, they were stored in sealed bags at room temperature until analysis. Commercial corn (*Zea mays* L.) starch (CS) and chemically modified starch (MS) were purchased from the local market and were used to compare their functional properties with those of the two extracted acorn starches. All chemicals and reagents were of analytical grade.

### 2.2. Starch Analysis

#### 2.2.1. Chemical Composition

Chemical composition of the extracted acorn starches was determined according to the AOAC methods for moisture, ash, protein, lipid, and carbohydrate contents [23]. The pH of acorn starch was determined using the method described by Achri et al. [24]. Color measurement of the studied samples was performed with a chromameter (Konika Minolta, CR-300, Tokyo, Japan) using the CIE LAB system (L*, a*, b*). The turbidity of starch suspensions was determined as described by Lan et al. [25] using a UV-visible spectrophotometer. A starch solution (1%, *w*/*v*) was prepared and then heated for 30 min in a boiling water bath with continuous stirring until starch gelatinization. After that, the starch was cooled for 30 min at room temperature. The light transmittance, expressed by the turbidity, of the gelatinized starches was measured by spectrophotometer at 620 nm, and distilled water was used as a blank [21].

#### 2.2.2. Solubility, Swelling Power, Water Absorption Capacity, Refrigeration, and Freezing Stability

Solubility, water absorption, and swelling power of different starch samples at 60, 70, 80, and 90 °C were determined according to the method described in the previous study by Zarroug et al. [13]. The evaluation of starch stability under refrigeration and freezing was examined also according to Zarroug et al. [13]. Briefly, starch solution (6% in water) was heated to 95 °C and then cooled to 50 °C by maintaining each treatment for 15 min. The syneresis rate was calculated after centrifugation of the solution at 4000 rpm for 10 min.

#### 2.2.3. Fourier Transform Infrared Spectroscopy

All spectra were collected using a PerkinElmer Spectrum single bounce ATR accessory with a diamond crystal. Dry starches were equilibrated at laboratory humidity (50% RH) and were dispersed in a matrix of KBr at room temperature. The obtained mixture was then pressed into pellets. The range of spectra was from 400 to 4000 cm^−1^ with a spectral resolution of 4 cm^−1^ [20].

#### 2.2.4. X-ray Diffraction (XRD)

X-ray diffraction (Bruker D8 Advance) in the Bragg–Bentano configuration with Ni-filtered Cu Kα radiation (k = 1.5418Å) operated at 40 kV and 40 mA was used to determine the degree of starch crystallinity according to the method reported by Zhang et al. [20]. Diffractograms were obtained in 2 h with a scanning speed of 2°/min and a scanning step of 0.02°.

#### 2.2.5. Thermogravimetric (TGA) Analysis

TGA is a technique where a mass loss of a sample can be measured as a function of time or temperature. Starch samples were analyzed by a thermogravimetric analyzer type Shimadzu TGA-50 (Shimadzu, Kyoto, Japan). Starch samples were heated from 20 to 500 °C in open alumina crucibles. A quantity of 8.0 mg of sample was used in a synthetic air stream of 100 mL/min with a heating rate of 10 °C/min [10].

#### 2.2.6. Thermal Properties

Thermal properties of starch samples were measured using a differential scanning calorimeter (DSC, Q2000, TA Co., New Castle, PA, USA). Starch sample (3 mg) was weighed into a DSC aluminum pan and distilled water (9 μL) was added with a micro syringe. The sample pans were hermetically sealed, reweighed, and allowed to stay overnight at room temperature in order to attain an even distribution of water before DSC analysis. Subsequently, the samples were heated from 20 to 120 °C at a rate of 10 °C/min, and an empty pan was used as a reference for all measurements. During the scans, the flow rate of dry nitrogen was kept at 50 mL/min. The gelatinization temperatures (T_0_, T_p_, and T_c_) and gelatinization (ΔH) were obtained from the DSC software [20].

#### 2.2.7. Scanning Electron Microscopy (SEM)

Acorn starch samples were dispersed evenly with a rubber suction bulb, placed on stub with double-sided adhesive tape, and coated with gold in an argon atmosphere. The surface of starch granules was analyzed under a scanning electron microscope (JSM–6360LV, JEOL, Tokyo, Japan) at an accelerating voltage of 20 kV.

### 2.3. Custard Preparation and Analysis

Custard recipe was prepared according to the following formula: 40 g of starch samples (WAS and AAS), 40 g sugar, 0.16 g vanilla flavor, and 500 mL whole fat milk. A commercial custard sample was also prepared in the same manner and used as control. For each custard sample, the ingredients were weighed into a glass and mixed with whole fat milk under mechanical stirring for 10 min and later warmed up to reach 100 °C. The mixture was kept at 100 °C for 4 min and subsequently cooled back until reaching room temperature. The prepared water acorn starch custard (WASC), alkaline acorn starch custard (AASC), and commercial custard (CC) were subjected to physicochemical (pH, moisture, protein, ash, fat, carbohydrate, and color parameter) analysis. In addition, the syneresis of custard samples was determined according to the method described by Ben Moussa et al. [26]. The microstructure of custard samples was also observed using scanning electron microscopy (SEM).

### 2.4. Statistical Analysis

All experiments were carried out in three replications, and the mean values ±STD were used for comparisons. Variance analysis (ANOVA) was performed using STATISTICA software to estimate significant differences between samples. Duncan’s multiple range tests was used for comparison at the significance level of 5% (*p* < 0.05).

## 3. Results and Discussion

### 3.1. Acorn Starch Characterization

#### 3.1.1. Physicochemical Characterization of Starches

In order to evaluate the quality of the extracted acorn starches using water (WAS) and alkaline (AAS) methods, their physicochemical properties were determined and compared to those of corn starch (CS) and modified starch (MS). Results of the physicochemical characterization of WAS, AAS, CS, and MS starch samples are shown in Table 1.

The yield values observed for WAS and AAS were about 48.32% and 48.1%, respectively. The starch yield from acorn *Quercus Suber* L. was higher than that reported in our previous study [13] on *Quercus ilex* L. starch (34.5%). In the same context, Irinislimane et Belhaneche Bensemr [27] and Correia et al. [28] found starch yields of 21% and 31.4% in Quercus *Suber* L., respectively. Previous studies have shown some variations in starch yield for starches extracted from bambara samples (values varied from 22 to 33%) [29] and cassava samples (values ranged from 67.36 to 81.13%) [30]. Significant differences at *p* ≤ 0.05 were observed in terms of ash, fat, carbohydrates, and protein values for the four studied starches, while no significant difference was found between WAS and AAS in terms of moisture content values. The moisture content values for WAS and AAS were 12.2% and 12.41%, respectively, and these values were closely related to the standard range recommended by codex *alimentarius* (<14%) for safe storage and minimal microbial growth leading to longer shelf life. Among all starches, the lowest moisture content was observed in the MS sample (7.95%). According to Kormin et al. [31], a moisture content lower than 10% is specified for incorporation into a low-density polyethylene matrix for the production of biodegradable products. The values for protein content in the studied starches were in the range of 0.12–0.30%, and the highest value was observed in the WAS sample. This result was lower than that found (0.92%) in the work on *Quercus ilex* L. starch [13]. The fat content in the studied starches ranged between 0.44% and 0.61%, and the highest fat content was noted in the AAS sample. These findings are in accordance with those obtained by Pérez et al. [21] and Shadrack et al. [30] on ramon (0.49%) and cassava (0.17%) starches, respectively. In general, the lipids and proteins in starch granules can increase their functionality. The protein in starch granules is associated with grain hardness, while the lipids can considerably reduce the swelling power of the starch paste. The ash contents in WAS and AAS was about 0.35% and 0.24%, respectively. These values are higher than those of commercial CS (0.17%) and MS (0.07%) samples. According to Zarroug et al. [13], the low protein, fat, and ash contents indicate high extracted starch purity and quality. The pH values for the starch samples ranged between 5.66 (WAS) and 6.81 units (MS). The obtained values for the extracted acorn starches (WAS and AAS) were lower than those found by Elmi et al. [32] in potato starch (6.22 unit). The AAS sample showed higher pH values when compared to the WAS sample. This variation was explained by the application of the NaOH concentration at 0.25M. According to Usman et al. [33], high pH could lead to undesirable protein modification as well as molecular cross linkage and rearrangements resulting in the formation of toxic compounds. A positive correlation was obtained between the pH value, the fat, and the protein content [34]. The color of starch samples has an impact on their quality and purity. It may also influence consumer preferences for the final prepared product. According to Franklin et al. [16], a high value for lightness (L* value) and a low value for chroma (a* and b*) are desired in starches. The extracted acorn starches (WAS and AAS) showed no significant difference (*p* ≤ 0.05) in L* and b* values, while, a significant difference was found between the a* values of these starches. Results revealed that commercial starches have higher L* values compared to the extracted acorn starch samples. However, among these extracted starches, the AAS showed a higher L* value (60.41) and lower a* (0.89) and b* (15.18) values when compared to the WAS sample. These findings were confirmed by the lower contents of protein and ash in AAS compared to WAS sample. In addition, the low L* values observed in the extracted acorn starches could be attributed to the presence of phenolic compounds in acorn flour. These phenolic compounds cannot be removed without oxidative modification, which eliminates the pigmented compounds by reaction with the appropriate reagent. Authors suggested that the relatively high L* values (greater than 90) and low a* and b* values of starches are indexes of purity [29]. In general, turbidity is used to characterize the retrogradation behavior of diluted starch paste [35]. The turbidity of the commercial starches (CS and MS) was lower than that of the acorn extracted starches (WAS and AAS) (Table 1). In addition, the WAS exhibited a higher turbidity value compared to AAS, because it contained higher protein and ash content. The high turbidity value was attributed to less complete swelling of starch granules, leading to the formation of a network between amylose and amylopectin chains that were leached out of the granules during gelatinization [36] and had a significant effect on the reflection or scattering of light.

#### 3.1.2. Swelling Power, Solubility, and Water Absorption of Starches

Results for the swelling power, solubility, and water absorption of different starches at temperatures ranging from 60 to 90 °C are presented in Table 2. The swelling power, solubility, and water absorption increased significantly (*p* < 0.05) from 60 to 90 °C among the studied starches, and the AAS sample showed higher values compared to the WAS sample. The observed swelling power values in AAS (6.05–22.51 g/g) and WAS (3.01–20.7 g/g) were higher compared to those of starch extracted from *Quercus fabri* Hance (4.7–13.5 g/g) but lower than those of *Quercus ilex* starch (4–20.76 g/g) [13] and sweet potato starch (35–40 g/g) [37]. The high swelling power of the starch samples may be attributed to an increase in the amorphous portion content [38], the starch molecule’s ability to hold water, and the degree of crystallinity [39]. The highest values for solubility and water absorption capacity, respectively, were found in the AAS sample at 90°C (13.3% and 14 g/g) followed by WAS (7.27% and 13.97 g/g), CS (6.15% and 8.31 g/g), and MS (4.34% and 5.8 g/g). These solubility values were lower than those reported by Jiang et al. [40] for starches extracted from five different *Dioscorea* L. species, ranging from 11.14 to 30.04% at the temperature of 95 °C.

The water absorption capacity corresponds to the integrity of starch in aqueous solutions and the volume of the formed gel, and it depends also on the availability of hydrophilic groups and on the capacity of the macromolecule to form a gel. The obtained water absorption results were higher than those reported by previous research [16] on *C. angustifolia* starch. These variations are due to many differences in the bonding strengths within the starch granules between amorphous and crystalline domains, which are influenced by granule starch characterization and the dissimilarity in amylose and amylopectin structure, including molecular weight distribution, chain length, degree/length of branching, and amylose–amylopectin ratio [41]. The reason for the higher solubility and water absorption capacity could be attributed to the ability of starch to absorb and retain water in the AAS sample and also to its crystalline molecular structure [20]. In fact, during the starch extraction at a high temperature, this crystalline structure may be broken and changed, causing the formation of bonds between the water molecules and the free hydroxyl groups of amylose and amylopectin chains. Authors suggested also that the method used for starch extraction may have an effect on the starch properties and may result in starches with different levels of surface-associated material, which may impact starch water absorption and swelling [39]. Indeed, Zhang et al. [20] reported that acorn starch isolated using the ultrasonic-assisted ethanol soaking method had relatively higher values for swelling power (24.99 g/100 g) and solubility (15.22%) at 90 °C compared to that obtained with other methods.

#### 3.1.3. Refrigeration and Freezing Stability of Starches

The stability variations in different studied starches under refrigeration and freezing conditions during storage times of 24, 48, 72, 96, and 120 h are shown in Table 3.

Results showed that the syneresis values at 4 °C and −20 °C of studied starches increased significantly (*p* < 0.05) with the increase in storage time. During 5 weeks of storage at 4 °C, the syneresis of acorn starch gels ranged from 42.4% to 62.99% and 41.82% to 46.07% for WAS and AAS, respectively, and the highest values were observed in the WAS sample. The same tendency was observed during freezing storage (−20 °C) for syneresis of the extracted acorn starches. In the current study, the syneresis values at −20 °C for the AAS sample were higher than those reported in a previous study [13] on *Quercus ilex* L. starch (34.5%). Several studies reported that the high level of amylose content in starches could be the major reason for the high syneresis. Starches extracted from potato, maize, taro, and cassava present high syneresis, due to the large amount of water expelled during the retrograding process [21,22,23,24,25,26,27,28,29,30].

#### 3.1.4. Thermal and Retrogradation Properties

A differential scanning calorimeter (DSC) was used to study the effect of thermal processing on the hydrated starch and to calculate the heat energy needed for starch gelatinization [42]. Thermal and retrogradation properties of different starches are summarized in Table 4. The To, Tp, and Tc temperature values for the studied starches ranged from 39.32 °C (MS) to 45.18 °C (AAS), 73.72 °C (MS) to 84.66 °C (AAS) and 115.35 °C (MS) to 118.92 °C (AAS), respectively. Results showed that the highest To, Tp, and Tc values were observed in AAS, followed by WAS, CS, and MS samples. We noted also that the extracted acorn starches (AAS and WAS) showed higher thermal temperatures and enthalpy values compared to other commercial starches (CS and MS).

From an industrial point of view, the onset temperature (T_0_), which represents the initiation of the gelatinization stage in starch granules, is associated with the costs of the used processes in terms of energy and time, and thus a low temperature is required by manufacturers.

Previous study on acorn starches revealed gelatinization temperatures and enthalpy values ranging from 55 to 75.2 °C and 6.52 to 20.8 J/g, respectively [10,43]. The greater gelatinization temperatures in acorn starches may be attributed to the longer double-helix chain of the crystal layer, which needs more energy to destroy the crystal structure compared to commercial starches. These findings are in accordance with previous studies on *Quercus ilex*, *Quercus suber*, and *Quercus rotundifolia* starches [10,39]. The thermal properties of the CS sample were similar to those found by Pérez-Pacheco et al. [21]. The gelatinization enthalpy (ΔHG) values for all studied starches ranged from 17.67 (WAS) to 23.40 (AAS) J/g. According to Boukhelkhal and Moulai-Mostefa, [10], the enthalpy of starch gelatinization is related to starch structure and molecular arrangement; usually, high enthalpy values are correlated to high organization of the amylopectin crystals. Furthermore, the amylose content could also influence the enthalpy of gelatinization since the studied starches were more organized and required higher energy to break hydrogen bonding. Therefore, the alkalis extracted acorn starch (AAS) retained a better molecular order than the WAS sample. In this study, the thermal properties differed from one type of starch to another, and these variations were due to several factors such as the difference in starch extraction methods, mineral composition, amylose/amylopectin ratio, starch granule morphology, starch structural arrangement, and botanical source, as well as the environmental conditions [10,11,12,13,14,15,16,17,18,19,20,21,22,23,24,25,26,27,28,29,30,31,32,33,34,35,36,37,38,39,40,41,42,43,44].

#### 3.1.5. X-ray Diffraction and FTIR Spectral Analysis

The molecular arrangement of the extracted starch granules was determined by X-ray diffraction and is shown in Figure 1a. The structural crystallinity of starch granules depends on their botanical source and is classified into three types of crystallinity patterns: A (Bragg angle 2θ at about 15.3°; 17.1°; 18.2° and 23.5°), B (Bragg angle 2θ at about 14.4°; 17.2°; 22.2° and 24.0°) and C (Bragg angle 2θ at approximately 15.3°; 17.3° and 23.5°) [30,31,32,33,34,35,36,37,38,39,40,41,42,43,44,45]. In general, most cereal starches are classified as type A. The arrowroot, cassava, and other plant tuber starches were reported to also have an A-type crystalline structure [45]. Native starches from various botanical sources are usually divided into A-, B- and C-type according to their crystallinity types. The studied CS and MS starches were X-ray A pattern crystals, with diffraction peaks corresponding to the emission angles (2θ) of 15.3°, 17.4°, 18°, 20°, and 23°. Similar findings were obtained by Ascheri [46] and Teixeira et al. [47] on Adlay (*Coix lacryma Jobi* L.) and waxy maize starches, respectively. However, the extracted acorn starches (WAS and AAS) showed strong reflections at 15.3° and 23° of 2θ. Results revealed that acorn starches are classified as type A. In addition, minor peaks were observed at 17.4° and 19.4° of 2θ, which are attributed to starches of type B. These results showed that acorn starches represent a mixture of type A and B crystallinity patterns. Similar results were reported by Zarroug et al. [13] for starch extracted from *Quercus ilex* L. The C-type crystalline structure consists of A- and B-type crystallites, so the X-ray diffraction pattern can contain various superpositions of the characteristic diffraction peaks depending on the ratio between the contents of these polymorphs [48]. Numerous other studies have already been conducted on the X-ray diffraction of acorn starches and showed a C-type [20,21,22,23,24,25,26,27,28,29,30,31,32,33,34,35,36,37,38,39,40,41,42,43]. The differences in the diffraction pattern of starch granules are mainly influenced by the genotypic, agronomic, and growing conditions of the source plant [49].

The FT-IR spectra of the starch samples are shown in Figure 1b. The observed spectra are similar for the four studied starches, with a broad intense band of 3600–3000 cm^−1^ representing the stretching vibrations of O–H bonds in the hydroxyl groups. A peak at 2931 cm^−1^ was attributed to the stretching vibration of the C–H bond in glucose, and the weakest peak intensity—reflecting the highest amylose content—was found in WAS followed by AAS, MS, and CS. According to Kizil, Irudayaraj, and Seetharaman [29,30,31,32,33,34,35,36,37,38,39,40,41,42,43,44,45,46,47,48,49,50] and Oyeyinka et al., the variation in peak intensity was explained by differences in amylose and amylopectin contents in corn, potato, wheat, and bambara starches. Additionally, significant bond peaks observed at 1566–1656 cm^−1^ were attributed to stretching vibrations of C–O bonds in the amide groups of proteins. The observed peaks at 1566, 1580, 1652, and 1656 cm^−1^ for WAS, CS, AAS, and MS, respectively, reflecting the O–H stretching bonds, were also assigned to absorbed water molecules (H_2_O) in the amorphous (non-crystalline) region [13,14,15,16,17,18,19,20,21,22,23,24,25,26,27,28,29,30,31,32,33,34,35,36,37,38,39,40,41,42,43,44,45,46,47,48,49,50]. The peaks found between 400 and 1162 cm^−1^ were effectively attributed to C–O bond stretching corresponding to vibration strain of the connections of C–O–C and C–O–H in β-glycosidic bonds [29]. These results are in agreement with those obtained for *Quercus ilex* L. [13], *S. lycocarpum* fruits [9], and Chinese ginkgo starches [51].

#### 3.1.6. Thermogravimetric (TGA) Analysis

The thermogram (TGA) and its derivative (Dr-TGA) of the studied starches as a function of temperature are plotted in Figure 2.

The different starches showed three stages of weight loss in the temperature range between 24.4 and 500 °C. The first stage of weight loss, observed between 24.40 and 141.73 °C, corresponded to the evaporation of water present in all starch samples. The highest weight loss was found in the WAS (17.98%) sample, followed by MS (16.39%), CS (15.04%), and AAS (10.29%) samples. After a stabilization stage, an important mass loss was observed between 278.80–332.03 °C (60.61%), 276–331.37 °C (58.22%), 237.61–327.61 °C (55.35 %), and 254.58–344.28 °C (49,53%) in MS, CS, AAS, and WAS, respectively. As shown, the extracted starch samples (WAS and AAS) exhibited lower mass loss compared to commercial starches (CS and MS). In this second stage, the major mass loss was due to the degradation and depolymerization of starch macromolecules (amylose and amylopectin). Moreover, mass loss values for the extracted acorn starches were lower than those reported by Boukhelkhal et al. [10] on acorns of holm oak (*Quercus ilex* subsp. ballota (Desf.) Samp.) grown in Algeria: HoS-BC sample (76.07%), HoSTG sample (65.07%), HoS-BG (69.76%), and HoS-YK (67%). The third stage was observed at temperatures between 327.61 and 500.79 °C and was the consequence of the carbonization of the inorganic components (fat and protein) and ash formation. Among all starches, the lowest mass loss was observed in the CS sample (10.51%), followed by MS (14.2%). These findings indicate the purity of the commercial starches. Results were in accordance with Liu et al. [52], who found in corn starch a reduction in mass of up to 8% in the temperature range between 150 °C and 70% in the temperature range from 250 to 360 °C. Hence, it could be stated that the commercial starches were thermally more stable than the extracted acorn starches.

#### 3.1.7. Microstructure of Starch Samples

Obtained results on various starch morphologies are shown in Figure 3. The studied starch granules showed different structures. When observed under SEM, the WAS and AAS granules (Figure 3a,b) were similar in structure and had a spherical and oval shape, with particle sizes ranging from 5 to 12 μm. These values are generally higher than those reported by Ali et al. [53] and Liu et al. [54] on rice starch (3.3–8.8 µm) and chestnut starch (1–16 μm), respectively. The surfaces of WAS granules appeared to be more smooth compared to those of AAS granules, which showed the presence of some pores and cracks. However, the CS and MS granules (Figure 3c,d) exhibited asymmetric, irregularly shaped granules with mainly ellipsoid, oval, and prismatic shapes. The particle sizes of CS, ranging from 3.5 to 24 μm, were similar to those reported by Hernández-Medina et al. [55] (10.6–33 µm) for corn and potato starches. The surfaces of the commercial starches appeared to be smooth, with no evidence of any fissures. Among all studied starches, the starch sample extracted by the alkaline method (AAS) showed the most damaged starch granules. This may be due to the use of alkaline solution (NaOH) for starch isolation, which caused damage to the amorphous area of the acorn starch, indicating lower resistance of acorn starch to alkali [20]. The variations in starch granule morphology may be due to the extraction and purification methods, environmental conditions, growth stage of the plant, and genotype [1]. In fact, the morphology of starch granules depends on the biochemistry of the chloroplasts or amyloplasts, as well as plant physiology [56]. The microstructural characteristics of acorn starch granules found in this study were similar to those reported by Zhang et al. [20].

### 3.2. Characterization of New Custard

#### 3.2.1. Physicochemical Composition of the Formulated Custard

In order to expand the use of extracted acorn starches at industrial scale, physicochemical properties and syneresis of the control and newly formulated custards were investigated, and the results are presented in Table 5.

Moisture values for prepared custard samples ranged from 76.76 to 77.77%, showing slight changes as a function of the starches used. In fact, polysaccharides are barriers to moisture due to their hygroscopic effect. Carbohydrates were the second major component of custard samples after moisture, with values ranging from 15.39 (AASC) to 18% (CC). The protein, ash, and fat contents in AASC and WASC samples were substantially higher than those in the control custard (CC). This finding can be explained by the fact that AASC and WASC samples were prepared from crude extracted acorn starches, whereas the CC was made from industrial pure corn powder. The protein and ash contents in formulated custards were lower than those obtained by Salami et al. [57] for corn starch custard, with values of 8.62% and 4.81%, respectively, for AASC and WASC samples. The pH of custard samples ranged from 6.64 to 6.99, and the highest value was observed in the AASC sample. The custards prepared with acorn starches presented higher pH values compared to the CC sample. A significant difference was observed in color parameters of the studied custard samples. The CC sample showed the highest L* (52.94) and brightness b* (34.67) values (Figure 4). The redness value a* ranged from 3.07 (CC) to 7.4 (WASC). These findings are similar to those reported in previous research [57]. Hence, the lower redness and higher brightness might be due to the interaction of other ingredients and the presence of carotenoids in the extracted starches.

#### 3.2.2. Syneresis of the Formulated Custard

The syneresis phenomenon is a major appearance-related issue in commercial custard that leads to an increase in whey liquid on the gel surface as a consequence of the reorganization of starch molecules (amylose and amylopectin), and can lower consumer acceptance of refrigerated commercial products [26]. The values for syneresis in different custard samples were in a range of 3.32–32.34%. As observed in Table 5, syneresis in all custard samples increased under refrigerated (4 °C) conditions with the increase in storage time. The CC sample showed lower syneresis values compared to the WASC and AASC samples throughout the storage period. The highest syneresis value was noted in the WASC prepared with acorn starch extracted by the water method. These results were in agreement with the findings of other studies on custard prepared with some fruits such as: apple powder [58], pineapple peel powder [59], and kiwi fruit marmalade [60]. The higher syneresis in custards prepared with acorn starches could be explained by the thermodynamic incompatibility between extracted starches and milk proteins. According to Sah et al. [59], the syneresis increase was attributed to the contraction of the casein network in the prepared custard samples.

#### 3.2.3. Microstructure of the Formulated Custard

In SEM micrographs (Figure 5), the different starch-based custard preparations showed a homogenous protein matrix where starch was uniformly distributed. In the network of the control custard, some cavities mainly attributed to the break in gel network structure appeared. These results showed that the microstructure of custards depended on starch nature as well as the dispersing phase type. In fact, Velez-Ruiz et al. [61] reported that the preparation of starch-based custard using milk resulted in a denser network when compared to the use of water as dispersing phase.

## 4. Conclusions

In this study, starch was extracted from acorn fruits (*Quercus suber* L.) using hot water soaking and alkaline washing methods. The extracted starches showed interesting physicochemical, functional, and structural properties when compared to commercial starches. The WAS and AAS acorn starches presented a considerable yield (48.32% and 48.1%). It was shown that commercial starches were lighter than the extracted acorn starches. Concerning functional characteristics, starches isolated from acorn fruits exhibited the highest swelling power, solubility, and water absorption, as well as greater enthalpy and gelatinization temperatures compared to commercial starches. Results from surface structure analysis showed that all studied starches had spherical and ovoid granules. However, acorn starches presented a C-type crystalline pattern, while commercial starches exhibited the A- type pattern. Determining the required characteristics of starch is necessary to select the best extraction method according to the application requirements, market trends, availability, structural characteristics, and cost. Moreover, the specific properties of underexploited *Quercus suber* L. starches encourage their use in innovative food products. Thus, new custards based on isolated acorn starches were prepared. The acorn starch-based custards exhibited interesting nutritional and functional properties and a homogenous protein matrix with uniformly distributed starch granules when compared to commercial custard.

## Figures and Tables

**Figure 1 polymers-14-00556-f001:**
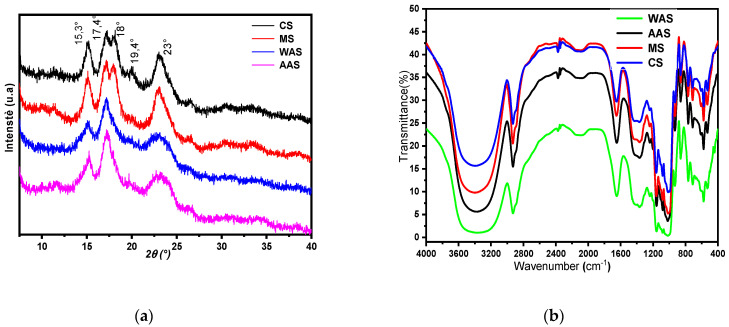
X-ray diffraction pattern (**a**) and FT-IR spectra (**b**) of different starches. WAS: water acorn starch, AAS: alkaline acorn starch, CS: corn starch, MS: modified starch.

**Figure 2 polymers-14-00556-f002:**
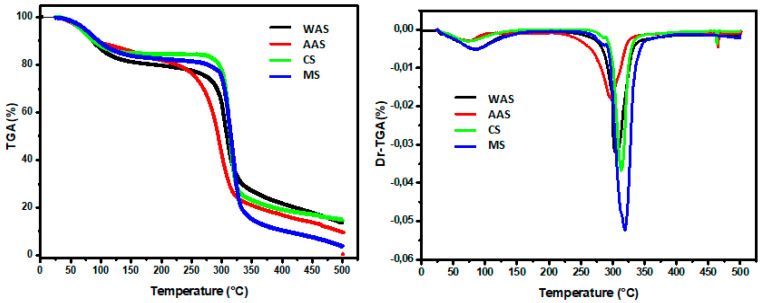
TGA and Dr-TGA curves for different starches. WAS: water acorn starch, AAS: alkaline acorn starch, CS: corn starch, MS: modified starch.

**Figure 3 polymers-14-00556-f003:**
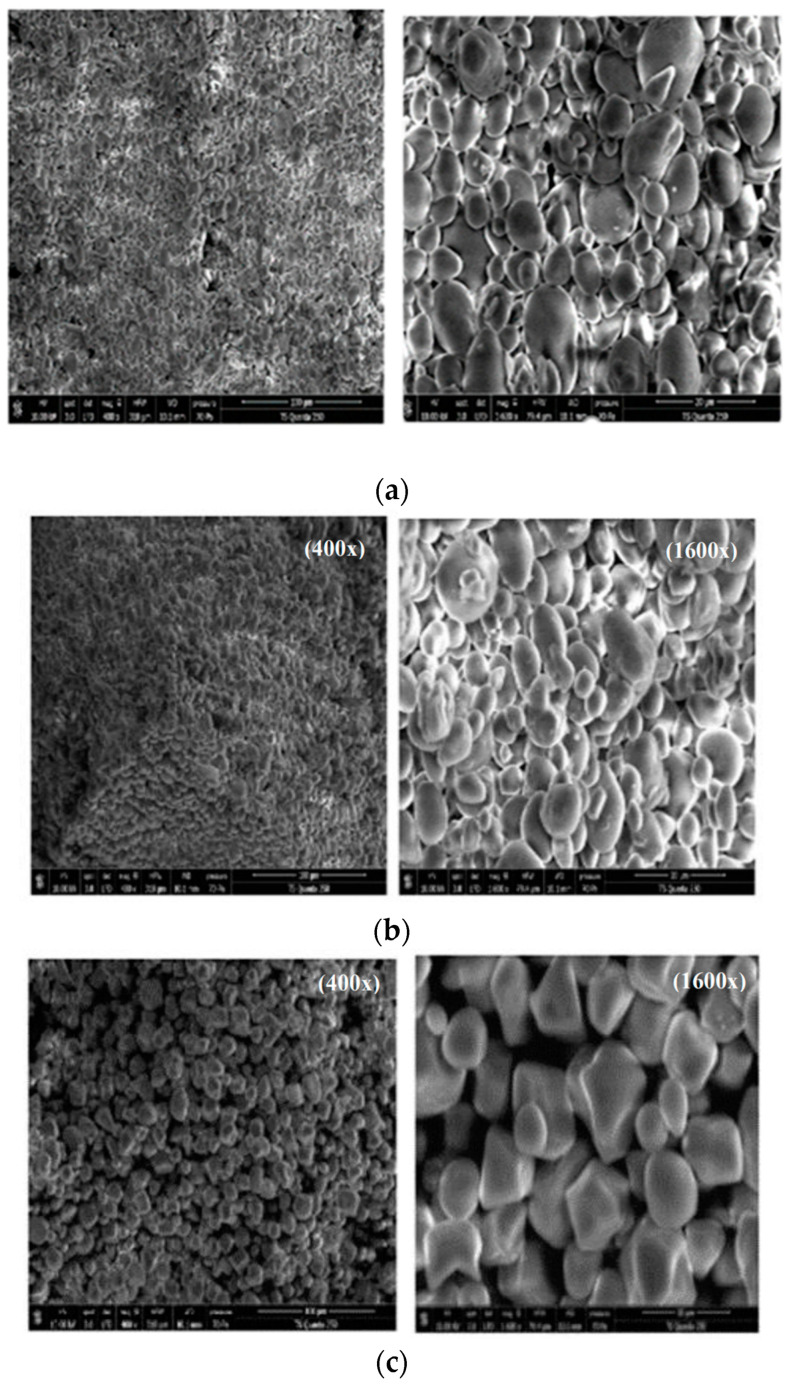
SEM images of different starches, magnification 400× and 1600×. (**a**) WAS: water acorn starch; (**b**) AAS: alkaline acorn starch; (**c**) CS: corn starch; (**d**) MS: modified starch.

**Figure 4 polymers-14-00556-f004:**
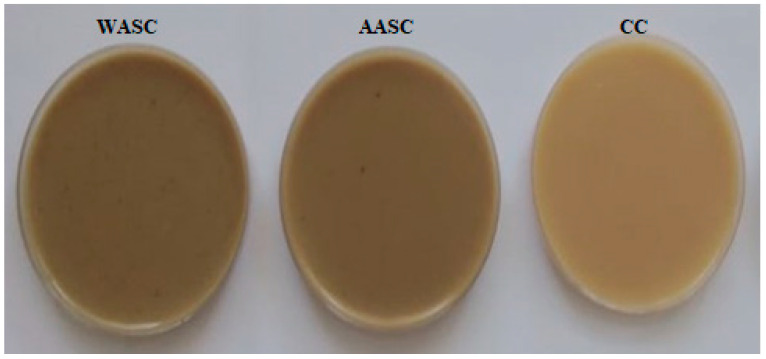
Appearance of different starch custards. CC: commercial custard, WASC: water acorn starch custard, AASC: alkaline acorn starch custard.

**Figure 5 polymers-14-00556-f005:**
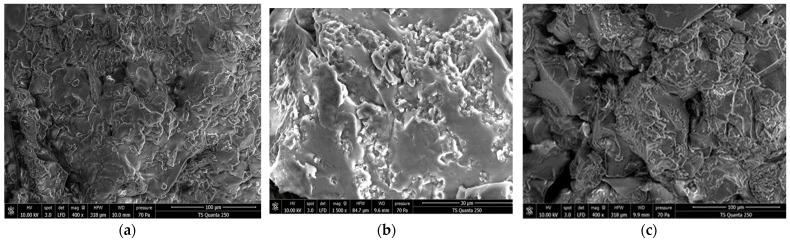
SEM images of different starch custards: (**a**) WASC: water acorn starch custard, (**b**) AASC: alkaline acorn starch custard, (**c**) CC: commercial custard. Magnification × 1600.

**Table 1 polymers-14-00556-t001:** Chemical composition of different starches.

Components	WAS	AAS	CS	MS
Yield (%)	48.32 ± 0.007 ^a^	48.1 ± 0.07 ^b^	-	-
Moisture (%)	12.20 ± 0.16 ^b^	12.41 ± 0.13 ^b^	9.16 ± 0.45 ^c^	7.95 ± 0.22 ^a^
Fat (%)	0.44 ± 0.007 ^a^	0.61 ± 0.028 ^d^	0.35 ± 0.014 ^b^	0.47 ± 0.01 ^c^
Protein (%)	0.30 ± 0.01 ^d^	0.27 ± 0.01 ^c^	0.14 ± 0.01 ^b^	0.12 ± 0.01 ^a^
Ash (%)	0.35 ± 0.02 ^c^	0.24 ± 0.04 ^b^	0.17 ± 0.005 ^d^	0.09 ± 0.02 ^a^
Carbohydrates (%)	86.71 ± 0.01 ^a^	86.47 ± 0.02 ^b^	90.18 ± 0.01 ^c^	91.37 ± 0.02 ^d^
pH	5.66 ± 0.05 ^b^	5.85 ± 0.01 ^a^	6.61 ± 0.02 ^c^	6.81 ± 0.01 ^d^
L*	55.76 ± 1.70 ^a^	60.41 ± 0.23 ^a^	87.8 ± 0.46 ^c^	92.72 ± 0.31 ^b^
a*	1.47 ± 0.234 ^d^	0.89 ± 0.02 ^c^	−0.27 ± 0.04 ^b^	−1.98 ± 0.12 ^a^
b*	15.81 ± 0.04 ^a^	15.18 ± 0.64 ^a^	−1.15 ± 0.13 ^a^	3.36 ± 0.22 ^a^
Transmittance (%)	56.10 ± 0.75 ^c^	40.80 ± 0.45 ^a^	15.67 ± 0.62 ^d^	23.67 ± 0.99 ^b^

Different letters in the same row indicate significantly different mean ± standard deviation of triplicates (*p* < 0.05). WAS: water acorn starch, AAS: alkalis acorn starch, CS: corn starch, MS: modified starch.

**Table 2 polymers-14-00556-t002:** Swelling power, solubility, and water absorption of different starches.

	Solubility (%)				Swelling Power (g Water/g Starch)				Water Absorption (g Water/g Starch)			
Samples	60 °C	70 °C	80 °C	90 °C	60 °C	70 °C	80 °C	90 °C	60 °C	70 °C	80 °C	90 °C
WAS	0.19 ± 0.01 ^a^	3.37 ± 0.01 ^c^	5.37 ± 0.02 ^c^	7.27 ± 0.01 ^c^	3.01 ± 0.01 ^a^	8.51 ± 0.01 ^c^	10.06 ± 0.01 ^c^	20.7 ± 0.01 ^c^	3.18 ± 0.01 ^c^	7.36 ± 0.01 ^c^	11.75 ± 0.01 ^d^	13.97 ± 0.01 ^c^
AAS	0.33 ± 0.01 ^c^	4.44 ± 0.01 ^d^	7.19 ± 0.01 ^d^	13.3 ± 0.14 ^d^	6.05 ± 0.02 ^d^	11.3 ± 0.14 ^d^	14.03 ± 0.01 ^a^	22.51 ± 0.01 ^d^	4.32 ± 0.02 ^d^	8.44 ± 0.01 ^d^	11.21 ± 0.01 ^c^	14 ± 0.01 ^d^
MS	1.89 ± 0.01 ^d^	0.14 ± 0.01 ^a^	0.47 ± 0.01 ^a^	4.34 ± 0.21 ^a^	4.83 ± 0.01 ^c^	3.17 ± 0.01 ^a^	3.36 ± 0.01 ^d^	3.68 ± 0.01 ^a^	1.88 ± 0.01 ^a^	4.14 ± 0.01 ^a^	5.34 ± 0.01 ^a^	5.8 ± 0.01 ^a^
CS	0.2 ± 0.01 ^b^	0.3 ± 0.01 ^b^	3.19 ± 0.01 ^b^	6.15 ± 0.00 ^b^	3.82 ± 0.01 ^b^	8.44 ± 0.08 ^b^	8.59 ± 0.01 ^b^	11.29 ± 0.01 ^b^	2 ± 0.01 ^b^	6.02 ± 0.01 ^b^	6.28 ± 0.02 ^b^	8.31 ± 0.01 ^b^

Different letters in the same row indicate significantly different mean ± standard deviation of triplicates (*p* < 0.05). WAS: water acorn starch, AAS: alkaline acorn starch, CS: corn starch, MS: modified starch.

**Table 3 polymers-14-00556-t003:** Refrigeration and freezing stability of different starches.

	Syneresis to Refrigeration at 4 °C (%)				Syneresis to Refrigeration at −20 °C (%)			
Time (h)	WAS	AAS	MS	CS	WAS	AAS	MS	CS
24	40.4 ± 0.01 ^b^	42.82 ± 0.01 ^c^	46.57 ± 0.01 ^d^	39.62 ± 0.01 ^a^	54.97 ± 0.01 ^c^	46.66 ± 0.01 ^b^	35.44 ± 0.01 ^a^	56.04 ± 0.01 ^d^
48	45.85 ± 0.01 ^c^	44.46 ± 0.01 ^b^	62.58 ± 0.01 ^d^	43.34 ± 0.01 ^a^	59.96 ± 0.01 ^d^	47.62 ± 0.01 ^c^	37.32 ± 0.01 ^a^	56.27 ± 0.01 ^b^
72	46.06 ± 0.01 ^c^	44.79 ± 0.01 ^b^	72.32 ± 0.01 ^d^	43.6 ± 0.01 ^a^	64.01 ± 0.01 ^a^	55.79 ± 0.01 ^c^	42.02 ± 0.01 ^b^	58.92 ± 0.01 ^d^
96	46.93 ± 0.01 ^b^	45.98 ± 0.02 ^a^	79.16 ± 0.01 ^d^	47.55 ± 0.01 ^c^	64.73 ± 0.01 ^d^	61.5 ± 0.01 ^b^	44.17 ± 0.01 ^a^	62.01 ± 0.01 ^c^
120	62.99 ± 0.03 ^b^	46.07 ± 0.01 ^a^	92.6 ± 0.01 ^d^	73.03 ± 0.01 ^c^	69.77 ± 0.01 ^d^	63.04 ± 0.01 ^c^	61.4 ± 0.01 ^a^	62.36 ± 0.01 ^b^

Different letters in the same row indicate significantly different mean ± standard deviation of triplicates (*p* < 0.05). WAS: water acorn starch, AAS: alkaline acorn starch, CS: corn starch, MS: modified starch.

**Table 4 polymers-14-00556-t004:** Thermal properties of different starches.

Samples	Thermal Properties				
	T_0_ (°C)	T_P_ (°C)	T_c_ (°C)	GR (°C)	ΔH_G_ (J/g)
AAS	45.18 ± 0.45 ^a^	84.66 ± 0.36 ^c^	118.92 ± 0.20 ^a^	78.96 ± 0.26 ^a^	23.40 ± 0.09 ^a^
WAS	41.67 ± 0.25 ^b^	78.36 ± 0.83 ^d^	117.41 ± 0.80 ^a^	75.38 ± 0.86 ^b^	17.67 ± 0.04 ^b^
CS	40.53 ± 0.60 ^b^	76.75 ± 0.40 ^b^	116.83 ± 0.7 ^a^	70.44 ± 0.91 ^c^	18.19 ± 0.27 ^c^
MS	39.32 ± 0.74 ^c^	73.72 ± 0.57 ^a^	115.35 ± 0.20 ^a^	68.80 ± 0.44 ^d^	19.79 ± 0.94 ^d^

Different letters in the same row indicate significantly different mean ± standard deviation of triplicates (*p* < 0.05). T_0_: onset temperature, T_P_: peak temperature, T_C_: conclusion temperature, ΔH_G_: enthalpy of gelatinization, GR: gelatinization range 2 (Tp–T_0_).

**Table 5 polymers-14-00556-t005:** Physicochemical properties, composition, and syneresis of acorn starch custards.

Samples	Moisture (%)	Fat (%)	Protein (%)	Ash (%)	Carbohydrates (%)	pH	L*	a*	b*	Syneresis (%)			
										24 h	48 h	72 h	96 h
CC	77.44 ± 0.27 ^c^	1.38 ± 0.01 ^a^	2.70 ± 0.01 ^a^	0.71 ± 0.06 ^a^	18 ± 0.01 ^c^	6.64 ± 0.02 ^a^	52.94 ± 0.02 ^c^	3.07 ± 0.01 ^a^	34.67 ± 0.01 ^c^	3.32 ± 0.01 ^a^	8.34 ± 0.01 ^b^	12.59 ± 0.01 ^b^	22.32 ± 0.01 ^a^
AASC	77.23 ± 0.11 ^b^	3.01 ± 0.01 ^c^	3.65 ± 0.01 ^c^	0.72 ± 0.03 ^a^	15.39 ± 0.01 ^a^	6.99 ± 0.02 ^c^	37.16 ± 0.01 ^b^	4.06 ± 0.01 ^b^	29.48 ± 0.01 ^a^	3.42 ± 0.01 ^b^	5.98 ± 0.01 ^a^	18.45 ± 0.01 ^a^	23.88 ± 0.01 ^b^
WASC	76.76 ± 0.64 ^a^	2.01 ± 0.01 ^b^	3.05 ± 0.01 ^b^	0.72 ± 0.03 ^a^	17.45 ± 0.01 ^b^	6.98 ± 0.01 ^b^	29.38 ± 0.01 ^a^	7.41 ± 0.01 ^c^	29.72 ± 0.01 ^b^	14.28 ± 0.01 ^c^	21.91 ± 0.01 ^c^	28.34 ± 0.01 ^c^	32.34 ± 0.01 ^c^

Different letters in the same row indicate significantly different mean ± standard deviation of triplicates (*p* < 0.05). CC: commercial custard, WASC: water acorn starch custard, AASC: alkaline acorn starch custard.

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
