# Peer review of "Structural and Physicochemical Properties of Tunisian Quercus suber L. Starches for Custard Formulation: A Comparative Study"

_polymers, 2022, doi:10.3390/polym14030556_

Round 1

Reviewer 1 Report

Youkabed Zrroung, et al. investigated structural and physicochemical properties of Tunisian Quercus suber. L starches for custard formulation. Basically, this is very good work. I have some suggestions for a minor revision.

  1. Please improve the resolution of Fig. 1 a, and Fig. 1b, particularly Fig. 1a. Looks like some information is missing. The width of lines in Fig. 1b needs to be increased.
  2. Please improve the quality of SEM images in Fig. 3.
  3. Please provide the scale bar in Fig. 4.
  4. In all legends, the authors need to state that these data are presentative or they are the average value of multimer experiments. The authors need to provide the statistic information in all legends.

Author Response

Youkabed ZARROUG

Field Crops Laboratory

National Agronomic Research Institute of Tunisia (INRAT)

E-mail: zarrougyoukabed@yahoo.fr

Tunis, 12 December, 2021

Dear Professor,

            Please find enclosed the revised manuscript entitled “Structural and physicochemical properties of Tunisian Quercus suber. L starches for custard formulation: A comparative study.” by Youkabed ZARROUG, Mouna BOULARES, Dorra SFAYHI, Bouthaina STITI, Bechir SLIMI, Kamel ZAYANI, Sirine NEFISI, and Mohamed KHARRAT. The original manuscript Reference is: Polymers-1486511.

We, the authors, have together considered all of the reviewer’s comments and made extensive corrections throughout the manuscript. Please find below our point-by-point answers to reviewer’s comments. Each reviewer is addressed individually, with the reviewer's comments in normal font and our answers in bold font. All changes in the revised manuscript have been amended by marking the text in red color.

I hope you will now consider the manuscript suitable for acceptance in Polymers journal.

Thank you and best regards.

Dr. Youkabed ZARROUG (on behalf of all the authors).

Field Crops Laboratory, National Agronomic Research Institute of Tunisia (INRAT)

E-mail: zarrougyoukabed@yahoo.fr

Reviewers' Comments to the Author:

Reviewer 1:

Comments and Suggestions for Authors

Youkabed Zrroung, et al. investigated structural and physicochemical properties of Tunisian Quercus suber. L starches for custard formulation. Basically, this is very good work. I have some suggestions for a minor revision.

  • Please improve the resolution of Fig. 1 a, and Fig. 1b, particularly Fig. 1a. Looks like some information is missing. The width of lines in Fig. 1b needs to be increased.

As suggested by the reviewer, the resolution of figures 1a and 1b were modified. The width of lines in the Fig.1b has been changed.

  • Please improve the quality of SEM images in Fig. 3.

As suggested by the reviewer, the quality of SEM images was improved.

  • Please provide the scale bar in Fig. 4.

As suggested by the reviewer the scale bar in Fig.4 was provided.

  • In all legends, the authors need to state that these data are presentative or they are the average value of multimer experiments. The authors need to provide the statistic information in all legends.

As suggested by the reviewer, the statistic information was added in all legends throughout the manuscript.

Reviewer 2 Report

The manuscript covers interesting topic on application of cork oak acorn starch for preparation of custard. The manuscript has proper composition and is prepared quite well. However the scope of the research is not perfectly clear to me – was the development of custard the main focus?. The first part of the research (characterization) is quite clear, nevertheless it would benefit from pasting analysis. The second part is in my opinion not complete it lacks sensory or at least texture evaluation. Moreover, it is not clear why custard prepared based on two other investigated starches used as a reference material were not included (CS and MS). The conclusions should be more detailed in terms of results and less explicit considering the applicability of AS (as its results are not that significant).

Detailed comments:

Line 73 – functional ingredient is often considered as functional food, please rephrase or provide detail why it possibly  could be considered as functional ingredient

Line 88 type of performed modification and botanical origin should be provided

Line 97 why Hunter L*a*b was used, currently CIE L*a*b is more common worldwide, especially in food industry

Line 98-102 please provide weather sample was cooled prior the measurement and for how long

Line 107-108 at least some detail should be provided

Line 116-120 were the starches conditioned prior the measurement?

Line 128-136 it was not staed how retrogradation was measured

Line 214-215 this sentence is quite crucial especially considering relatively low values of L, and should be discussed more in detail.

Line 215-216 while the statement is true, it is not directly linked with this study. For retrogradation measurements of changes turbidity over time a required

Line 218-220 please rephrase, high turbidity – less complete swelling

Line 246-254 during isolation time and temperature may have effect on annealing of the starch

Line 303-304 it is not that obvious: while cassava has C-type another tuber i.e. potato has B type crystalline structure

Figure 1a – presenting diffraction pattern of starch up angle 70 is pointless, and caused the graphs to be less readable, please cut the x-axis to at least 40°

Author Response

Youkabed ZARROUG

Field Crops Laboratory

National Agronomic Research Institute of Tunisia (INRAT)

E-mail: zarrougyoukabed@yahoo.fr

Tunis, 12 December, 2021

Dear Professor,

            Please find enclosed the revised manuscript entitled “Structural and physicochemical properties of Tunisian Quercus suber. L starches for custard formulation: A comparative study.” by Youkabed ZARROUG, Mouna BOULARES, Dorra SFAYHI, Bouthaina STITI, Bechir SLIMI, Kamel ZAYANI, Sirine NEFISI, and Mohamed KHARRAT. The original manuscript Reference is: Polymers-1486511.

We, the authors, have together considered all of the reviewer’s comments and made extensive corrections throughout the manuscript. Please find below our point-by-point answers to reviewer’s comments. Each reviewer is addressed individually, with the reviewer's comments in normal font and our answers in bold font. All changes in the revised manuscript have been amended by marking the text in red color.

I hope you will now consider the manuscript suitable for acceptance in Polymers journal.

Thank you and best regards.

Dr. Youkabed ZARROUG (on behalf of all the authors).

Field Crops Laboratory, National Agronomic Research Institute of Tunisia (INRAT)

E-mail: zarrougyoukabed@yahoo.fr

Reviewer 2

Comments and Suggestions for Authors

The manuscript covers interesting topic on application of cork oak acorn starch for preparation of custard. The manuscript has proper composition and is prepared quite well. However the scope of the research is not perfectly clear to me – was the development of custard the main focus?. The first part of the research (characterization) is quite clear, nevertheless it would benefit from pasting analysis. The second part is in my opinion not complete it lacks sensory or at least texture evaluation. Moreover, it is not clear why custard prepared based on two other investigated starches used as a reference material were not included (CS and MS). The conclusions should be more detailed in terms of results and less explicit considering the applicability of AS (as its results are not that significant).

As suggested by the reviewer, the objective of this study was the isolation and characterization of acorn starch. The comparative study performed between the quality of acorn starch and those of commercial starches was, then, performed in order to determine its potential use in food industry. Thus, the formulation of the new custard was the first and preliminary work to confirm the interesting technological properties of this new extracted starch. 

Concerning the pasting analysis, we agree with you, in fact, pasting analysis will be the next step of our study in our laboratory.

We agree with you but, as mentioned before, we focused during custard formulation on its microstructure and some properties. All other missing analyses will be performed in our next work because it will be more interesting to perform the evolution of all characteristics during storage period. Furthermore, initial obtained results on color parameters and syneresis level will give us idea about the quality of the final product particularly some sensorial and rheological properties.

Moreover, in this study, we compared the quality of acorn starch based custard to that of a commercial one prepared with corn starch (CS) used in the first part. Detail is provided in the material and methods part of the manuscript.

Detailed comments:

  • Line 73: functional ingredient is often considered as functional food, please rephrase or provide detail why it possibly could be considered as functional ingredient.

As suggested by the reviewer, the sentence was reformulated as requested. In fact, acorn flour is considered as functional product by providing a better antioxidant activity due to its richness in natural bioactive compounds. However, acorn starch improves rheological and textural properties of the final product.

The new sentence is:

“However, to our knowledge, the use of acorn starch for custard formulation has not been reported which could give importance to the production of custard using this new acorn starch.”

  • Line 88: type of performed modification and botanical origin should be provided.

As suggested by the reviewer, a chemically modified starch (MS) and commercial corn (Zea mays L.) starch (CS) were used in this study.

  • Line 97: why Hunter L*a*b was used, currently CIE L*a*b is more common worldwide, especially in food industry

As suggested by the reviewer, it was just a mistake. The used instrument was a CIE LAB scale.

The new sentence is:

Color measurement of studied samples was determined by a chromameter (Konika Minolta, CR-300, Japan) using the CIE LAB system (L*, a*, b*).”

  • Line 98-102: please provide weather sample was cooled prior the measurement and for how long

As suggested by the reviewer, starch was cooled for 30 min at room temperature.

The new sentence is:

“After that, the starch was cooled for 30 min at room temperature.”

  • Line 107-108: at least some detail should be provided

As suggested by the reviewer, some details were provided in the manuscript.

The added sentence is:

“Briefly, starch solution (6% in water) was heated to 95°C and then, cooled to 50°C by maintaining each treatment during 15 min. Syneresis rate was calculated after centrifugation of the solution at 4000 rpm for 10 min.”

  • Line 116-120: where the starches conditioned prior the measurement?

As suggested by the reviewer, the extracted acorn starches were dried and stored in sealed plastic bags at room temperature until analysis.

The added sentence is:

“The obtained starches were dried at 40°C for 12 h and milled into a fine powder. Then, they were stored in sealed bags at room temperature until analysis.”

  • Line 128-136: it was not stated how retrogradation was measured

As suggested by the reviewer, the sub-title was changed; it is a determination of the thermal properties of the starch samples. The results of thermal parameters (T0, Tp and Tc and ΔH) were obtained from the DSC software.

The new title is:

“2.2.6. Thermal properties”

  • Line 214-215: this sentence is quite crucial especially considering relatively low values of L, and should be discussed more in detail.

As suggested by the reviewer, the L value performed in our study on acorn starches was attributed to the initial color of acorn starch.

  • Line 215-216: while the statement is true, it is not directly linked with this study. For retrogradation measurements of changes turbidity over time a required.

The high turbidity value was attributed to less complete swelling of starch granules leading to formation of a network between amylose and amylopectin chains that were leached out of the granules during gelatinization [36] and had so a significant effect on the reflection or scattering of light.

  • Line 218-220: please rephrase, high turbidity – less complete swelling

As suggested by the reviewer, the sentence was reformulated.

The new sentences are:

“In addition, the observed low L* values in the extracted acorn starches could be attributed to the presence of phenolic compounds in acorn flour. These phenolic compounds cannot be removed without oxidative modification, which eliminates the pigmented compounds by reaction with the adequate reagent. Authors suggested that the relatively high L* values (greater than 90) and low a* and b* values of starches are an index of purity [29].”

“The high turbidity value was attributed to less complete swelling of starch granules and formation of a network between amylose and amylopectin chains that were leached out of the granules during gelatinization [36].”

  • Line 246-254: during isolation time and temperature may have effect on annealing of the starch

As suggested by the reviewer, the isolation conditions affect the absorption and swelling behaviors of the extracted starches.

A new sentence was added:

“In fact, during the starch extraction at a high temperature, this crystalline structure may be broken and changed that caused the formation of bonds between the water molecules and the free hydroxyl groups of amylose and amylopectin chains.”

  • Line 303-304: it is not that obvious: while cassava has C-type another tuber i.e. potato has B type crystalline structure

As suggested by the reviewer, more details are given in the manuscript.

These sentences are added:

“The starch granules structural crystallinity depending on their botanical source, have been classified into three types of crystallinity patterns: A (Bragg angle 2θ at about 15.3°; 17.1°; 18.2° and 23.5°), B (Bragg angle 2θ at about 14.4°; 17.2°; 22.2° and 24.0°) and C (Bragg angle 2θ at approximately 15.3°; 17.3° and 23.5°) [30-45]. “

“Native starches from various botanical sources are usually divided into A-, B- and C-type according to their crystallinity types.”

  • Figure 1a – presenting diffraction pattern of starch up angle 70 is pointless, and caused the graphs to be less readable, please cut the x-axis to at least 40°.

As suggested by the reviewer, the figure 1a was changed.

Reviewer 3 Report

The aim of the present research was to extract starch from acorn (Quercus Suber. L) fruits by water and alkaline methods.  Their structural and technological properties were investigated and compared to those of corn and modified starches. The isolated starches showed low moisture, fat and proteins contents (high purity and quality). Also, the extracted starch using alkaline method (AAS) showed higher lightness in comparison to starch isolated by hot water (WAS), but the lightest white color was found for studied commercial starches. AAS starch exhibited the highest swelling power, solubility and water absorption followed by WAS and commercial starches. The extracted acorn starches were characterized by greater enthalpy and gelatinization temperatures. Also, similar observations were noted using FT-IR spectra analysis for all analyzed starches. Besides, observed granule starches were found to be spherical and ovoid by scanning electron microscopy. In order to valorize the extracted acorn starches, they served to produce new custards with high nutritional properties and better microstructure when compared to commercial custard. The research has been performed adequately with appropriate methods. I have some minor remarks written further, and the important remark is related to the novelty of performed research.

Minor remarks:

The English grammar could be improved and it should be rechecked in overall manuscript.

In the abstract there is need to state what was done for the first time.

What is C pattern in the abstract (line 27) as well as A pattern (line 28)?

Lines 64-66: The sentence „Although several studies have examined the different methods used for starch extraction such as alcohols-based extraction, alkaline-based extraction and acetone-based extraction [17-18-19].“ is not clear and it should be rephrased.

At the end of introduction part there is need to state more clearly what is the novelty of present research. Is this the first report of Quercus suber. L starches or their uses? Briefly survey of available references revels that there are papers presenting starch isolation from Q. suber, like “The effect of starch isolation method on physical and functional properties of Portuguese nut starches. II. Q. rotundifolia Lam. and Q. suber Lam. acorns starches“, Food Hydrocolloids 30, 2013, 448-455. Therefore, there is need to point out more what was done for the first time in present research in comparison to previously published own research as well as from other authors.

The discussion part should be enriched with the data about the novelty of the present research in comparison with others.

Also in the conclusion part there is need to add the overall novelty of performed research.

Author Response

Youkabed ZARROUG

Field Crops Laboratory

National Agronomic Research Institute of Tunisia (INRAT)

E-mail: zarrougyoukabed@yahoo.fr

Tunis, 12 December, 2021

Dear Professor,

            Please find enclosed the revised manuscript entitled “Structural and physicochemical properties of Tunisian Quercus suber. L starches for custard formulation: A comparative study.” by Youkabed ZARROUG, Mouna BOULARES, Dorra SFAYHI, Bouthaina STITI, Bechir SLIMI, Kamel ZAYANI, Sirine NEFISI, and Mohamed KHARRAT. The original manuscript Reference is: Polymers-1486511.

We, the authors, have together considered all of the reviewer’s comments and made extensive corrections throughout the manuscript. Please find below our point-by-point answers to reviewer’s comments. Each reviewer is addressed individually, with the reviewer's comments in normal font and our answers in bold font. All changes in the revised manuscript have been amended by marking the text in red color.

I hope you will now consider the manuscript suitable for acceptance in Polymers journal.

Thank you and best regards.

Dr. Youkabed ZARROUG (on behalf of all the authors).

Field Crops Laboratory, National Agronomic Research Institute of Tunisia (INRAT)

E-mail: zarrougyoukabed@yahoo.fr

Reviewer 3

Comments and Suggestions for Authors

The aim of the present research was to extract starch from acorn (Quercus Suber. L) fruits by water and alkaline methods.  Their structural and technological properties were investigated and compared to those of corn and modified starches. The isolated starches showed low moisture, fat and proteins contents (high purity and quality). Also, the extracted starch using alkaline method (AAS) showed higher lightness in comparison to starch isolated by hot water (WAS), but the lightest white color was found for studied commercial starches. AAS starch exhibited the highest swelling power, solubility and water absorption followed by WAS and commercial starches. The extracted acorn starches were characterized by greater enthalpy and gelatinization temperatures. Also, similar observations were noted using FT-IR spectra analysis for all analyzed starches. Besides, observed granule starches were found to be spherical and ovoid by scanning electron microscopy. In order to valorize the extracted acorn starches, they served to produce new custards with high nutritional properties and better microstructure when compared to commercial custard. The research has been performed adequately with appropriate methods. I have some minor remarks written further, and the important remark is related to the novelty of performed research.

Minor remarks:

  • The English grammar could be improved and it should be rechecked in overall manuscript.

As suggested by the reviewer, the English grammar was improved throughout the manuscript.

  • In the abstract there is need to state what was done for the first time.

As suggested by the reviewer, to our knowledge, it is the first study in which extracted acorn starch was used for the formulation of new custard.

The new sentences in the abstract are:

“Structural and technological properties of extracted starches were investigated and compared to those of corn and modified starches in order to determine their innovative potential application in food industry.”

“However, from the analysis by X-ray diffraction, crystalline pattern of C-type was found for acorn starches while commercial starches presented A- type pattern. As innovative food application, these underexploited acorn starches were valorized and served to produce new custards with high technological properties and better microstructure when compared to commercial custard.”

  • What is C pattern in the abstract (line 27) as well as A pattern (line 28)?

As suggested by the reviewer, the sentence was corrected. In fact, the starch granules structural crystallinity depending on their source, have been classified into three types of crystallinity patterns (A, B and C).

The new sentence is:

“However, from the analysis by X-ray diffraction, crystalline pattern of C-type was found for acorn starches while commercial starches presented A- type pattern.”

  • Lines 64-66: The sentence „Although several studies have examined the different methods used for starch extraction such as alcohols-based extraction, alkaline-based extraction and acetone-based extraction [17-18-19].“ is not clear and it should be rephrased.

As suggested by the reviewer, the sentence was rephrased.

The new sentence is:

“For the acorn extraction starch, many chemical-based technologies have been extensively applied in research such as alcohols-based extraction, alkaline-based extraction and acetone-based extraction [17-18-19]. However, Zhang et al. [20] studied the effect of both physical and chemical methods, including hot-water soaking, alkaline washing, ultrasonic-assisted ethanol soaking and ultrasonic-assisted hot-water soaking, on the extracted acorn starch yield”.

  • At the end of introduction part there is need to state more clearly what is the novelty of present research. Is this the first report of Quercus suber. L starches or their uses? Briefly survey of available references revels that there are papers presenting starch isolation from Q. suber, like “The effect of starch isolation method on physical and functional properties of Portuguese nut starches. II. Q. rotundifolia Lam. and Q. suber Lam. acorns starches“, Food Hydrocolloids 30, 2013  Therefore, there is need to point out more what was done for the first time in present research in comparison to previously published own research as well as from other authors.

As suggested by the reviewer, many details were added to the introduction part.

The new added sentences are:

“These native starches of different botanical origins reveal enormous diversity in their structure and composition, including their amylase, amylopectin and minor constituent (lipids, proteins, and minerals) contents. Starch is rarely consumed in its intact form and frequently used by industry in its native form. Most native starches are limited in their direct application because they are unstable with respect to changes in temperature, pH and shear forces. Therefore, native starches are often modified to enhance their technological and functional properties under high heating temperatures used in industrial processes.”

“In food industry, starch was used, particularly, for improvement of baking flour and bakery products properties such as cakes and breads due to their high consumption. During their formulations, starch is one of the components responsible for a broad technological functionality. Generally, starch is used as food additive, as a hydrocolloid, thickener, emulsifier and coating agent [16] in many industrial foods to improve their technological and sensorial characteristics”.

“For the acorn extraction starch, many chemical-based technologies have been extensively applied in research such as alcohols-based extraction, alkaline-based extraction and acetone-based extraction [17-18-19]. However, Zhang et al. [20] studied the effect of both physical and chemical methods, including hot-water soaking, alkaline washing, ultrasonic-assisted ethanol soaking and ultrasonic-assisted hot-water soaking, on the extracted acorn starch yield.”

“Acorns are non-conventional and good sources of new form of starch that was not valorized until now in food industry. In fact, few studies focused on technological properties of acorn starch from Quercus species without any food applications. The first application of acorn starch was studied in the investigation of Zarroug et al. [13] who incorporated acorn starch during the production of a fermented dairy product to enhance its technological properties. To our knowledge, the use of acorn starch for custard formulation has not been reported before which show the importance of this innovative application aiming to valorize this underexploited raw material in food industry.”

“Second, the study aimed to investigate some properties of new formulated acorn-based custard to provide information on the potential incorporation of acorn starch in various industrial food applications.”

  • The discussion part should be enriched with the data about the novelty of the present research in comparison with others.

As suggested by the reviewer, the discussion part was enriched.

The new added sentences are:

“In general, lipids and proteins in starch granules can increase their functionality. The protein in starch granules is associated with grain hardness, while the lipids can considerably reduce the swelling power of the starch paste.”

“For an industrial point of view, the onset temperature (To), which represents the initiation of the gelatinization period of starch granules, is associated to the costs of the used processes regarding energy and time, and thus a low temperature is required by manufacturers.”

“In fact, polysaccharides are barriers to moisture due to their hygroscopic effect.”

  • Also in the conclusion part there is need to add the overall novelty of performed research.

As suggested by the reviewer, many details were added to the conclusion part.

The new added sentences are:

“Determining the required characteristics of starch is necessary to select the best extraction method according to the application requirements, market trends, availability, structural characteristics and cost. Moreover, the specific properties of underexploited Quercus suber. L starches encourage their use in innovative food products. Thus, new custards were prepared based on isolated acorn starches. The acorn starch-based custards exhibited interesting nutritional and technological properties and homogenous protein matrix having uniformly distributed starch granules when compared to commercial custard.”

Round 2

Reviewer 2 Report

The manuscript has been revised properly. In my opinion it would benefit from more information on characterization of acorn starch, nevertheless it is suitable for publication in current form.

Few corrections that should be done prior publication:

Line 44 – spelling mistake it is amylase, should be amylose

Line 51-52 style

Line 78 – type of chemical modification should be provided

Line 242 – for industrial application viscograph analysis is performed, in case of starch DSC is mainly for research purposes as it can be shifted with the macroscopic data

Line 399 – lightest is not clear in this case, please rephrase this sentence to be more precise

Line 402 – only corn starch was tested

Line 404- as above

Line 410 – nutritional aspects were not evaluated